# Blockchain-Powered Education: A Sustainable Approach for Secured and Connected University Systems

Van Duy Tran [1], Shingo Ata [1], Thi Hong Tran [1,*], Duc Khai Lam [2,3] and Hoai Luan Pham [2,3]

1 Graduate School of Informatics, Osaka Metropolitan University, 3-3-138 Sugimoto, Sumiyoshi-ku, Osaka 558-8585, Japan

2 Computer Engineering Department, University of Information Technology, Ho Chi Minh City 700000, Vietnam; khaild@uit.edu.vn (D.K.L.)

3 Vietnam National University, Ho Chi Minh City 700000, Vietnam

* Correspondence: hong@omu.ac.jp

**Abstract:** The collection and examination of student data, encompassing academic achievements, awards, and certifications, assume an essential function within the field of education as a means of showing students' capabilities. Nevertheless, it is crucial to note that regular paper-based records are vulnerable to both physical destruction and the act of fabrication, while standard databases can have security holes. Moreover, the process of manually gathering physical papers from centralized organizations is both laborious and complicated. To address the concerns above and foster sustainability in the field of education, this study first suggests using Scorechain. This innovative solution integrates blockchain technology into a comprehensive data-management system for managing all student-related data. Secondly, by utilizing the inherent security features of blockchain technology, Scorechain develops a stable multi-role hierarchy, increasing the integrity and reliability of data. This also facilitates the efficient transfer of information among various stakeholders, including parents, recruiters, and educational institutions, thus fostering transparency and accountability. Lastly, the Scorechain system facilitates collaboration and data exchange among universities inside a shared network. Scorechain was constructed using the Rust programming language and is based on the Substrate blockchain architecture. It underwent careful development, testing, and analysis to ensure operational efficiency. The feasibility and long-term viability of Scorechain in genuine educational contexts are highlighted as blockchain technology facilitates seamless integration into the education sector.

**Keywords:** blockchain; smart campus; Education 4.0; sustainable education

## 1. Introduction

In today's society, academic credentials are highly valued as they are a concrete indicator of an individual's human capital, encompassing his/her skills, knowledge, talents, and educational accomplishments [1]. In recent years, there has been an increasing concern regarding the widespread occurrence of fraudulent academic credentials globally. This tendency is frequently attributed to economic challenges, leading individuals to forge documents to obtain job prospects [2–4]. The conventional paper-based formats for storing student data, such as transcripts, certificates, and awards, are susceptible to counterfeiting and require strict verification protocols [5,6]. In addition, the process of manually verifying these documents imposes considerable strain on colleges and universities, resulting in the consumption of valuable time and resources. Despite the shift towards digital records, ongoing difficulties persist in data sharing among universities and various stakeholders, such as parents and recruiters. These challenges include the vulnerability to cyberattacks and concerns surrounding centralized data storage. These concerns encompass potential risks to security and privacy, as well as the associated financial costs.

The concept of blockchain technology was first introduced by Satoshi Nakamoto in 2008, alongside the creation of the Bitcoin cryptocurrency [7]. Over time, blockchain has developed into a decentralized, unchangeable, and transparent system for recording transactions and monitoring assets in diverse sectors such as finance, healthcare, education, transportation, and the Internet of Things (IoT).

The educational sector has experienced notable progress in implementing blockchain technology [8,9]. Several initiatives have been proposed in the academic domain, including the implementation of blockchain technology for storing student transcript information [10], the issue of verifiable academic certificates [11], the prevention of exam question leaks [12], the management of certificates for performance evaluation during recruitment [13], how to create an Ethereum blockchain-based project that is integrated with Moodle [14], and an Ethereum-based system to issue and manage academic certificates [15]. Nevertheless, these efforts frequently concentrate on discrete applications and fail to offer an integrated structure that can replace current university management systems. Furthermore, the extent of integration and data sharing among educational institutions remains constrained, affecting data privacy and collaboration.

This paper presents Scorechain as a comprehensive management system that leverages blockchain technology's decentralized, immutable, and transparent characteristics to address the above-mentioned difficulties. Scorechain surpasses traditional blockchain applications by securely storing student data, implementing a robust multi-role hierarchy system to enhance security and reliability, streamlining information communication to relevant parties, empowering individuals to manage their data autonomously, and facilitating smooth data transfer among universities.

The remaining sections of this paper are organized in the following manner: Section 2 offers a comprehensive analysis of blockchain systems. Section 3 provides a comprehensive analysis of the complicated components of the Scorechain system, while Section 4 focuses on the practical aspects of its implementation. In Section 5, an examination is conducted on the fundamental attributes and results of the Scorechain system. In conclusion, our study is brought to a close in Section 6, and we put out potential directions for further research.

## 2. Blockchain

### 2.1. Overview

Blockchain technology is a critical technological framework that enables the secure storage and transmission of data or money. Created as the foundational technology underlying Bitcoin, blockchain has subsequently been utilized in many fields, such as voting systems, supply chain management, and digital identity verification. The operations of blockchain involve the establishment of a secure and encoded ledger, including discrete data blocks. Each block encompasses transaction details, a timestamp, and a cryptographic hash derived from the previous block. This process ensures the creation of an unalterable and impervious record, safeguarding all network transactions. The maintenance of this network is carried out by a decentralized group of users referred to as nodes. Each node owns a complete copy of the blockchain and is responsible for validating network transactions. The general structure of a blockchain is depicted in Figure 1.

Decentralization, immutability, transparency, security, efficiency, traceability, and programmability are the characteristics described in more detail below and in Figure 2, making blockchain so attractive. Its decentralization implies that it is administered by a network of independent nodes, rather than a single entity. Immutability assures that, once a block is added to the blockchain, its contents cannot be modified or removed, resulting in a tamper-proof ledger of all transactions. Transparency allows anyone with network access to view the data, promoting accountability and transparency. Security is accomplished using cryptographic methods to encrypt transactions and avoid corruption and manipulation. At the same time, efficiency is gained by using a decentralized network that can process transactions and validate blocks rapidly. Traceability permits the tracking and verification of

every transaction, while programmability enables the creation of decentralized applications (dApps) that may automate complex tasks.

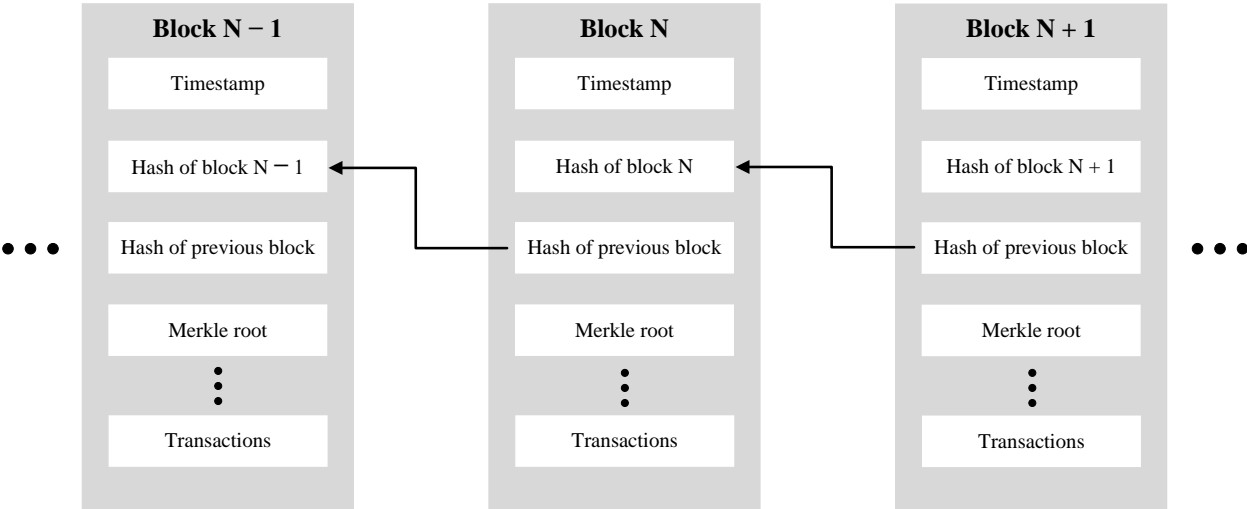

**Figure 1.** The general structure of the blockchain system.

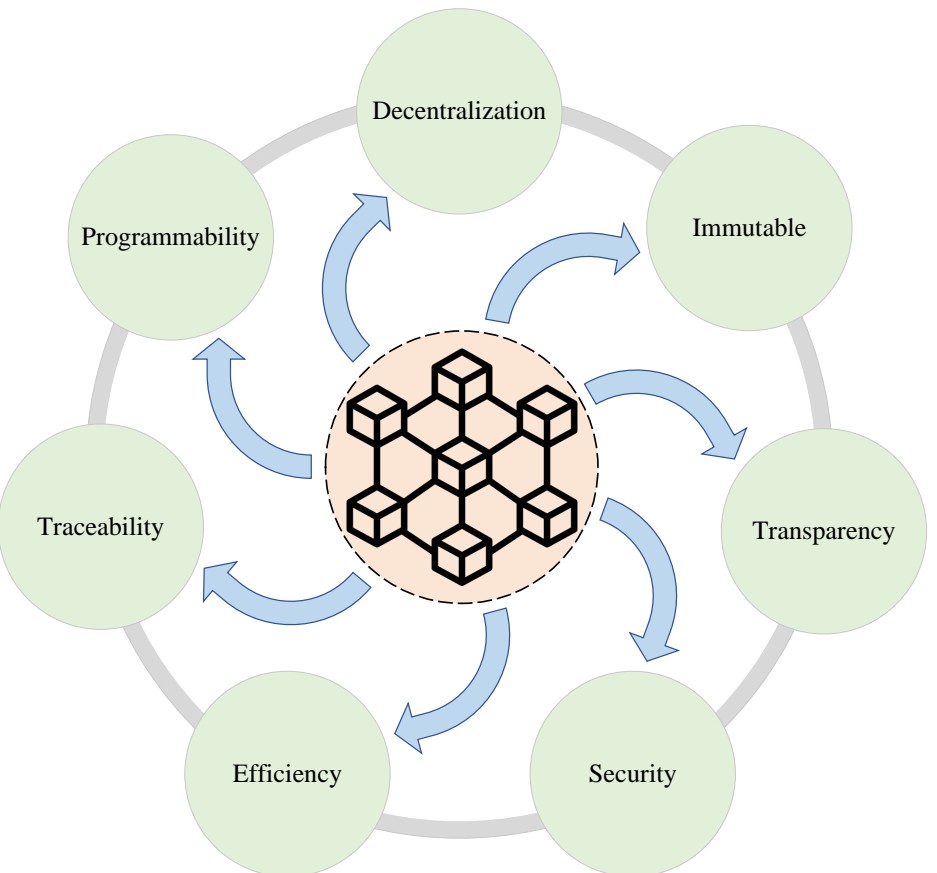

**Figure 2.** The main features of a blockchain system.

Figure 2 shows the essential qualities that make blockchain technology attractive, including decentralization, immutability, transparency, security, efficiency, traceability, and programmability. Decentralization refers to the operational structure of a network of independent nodes, avoiding the need for a central governing organization. The concept of immutability guarantees that, once data are appended to the blockchain, they become

highly resistant to any alteration or removal, safeguarding the integrity of all documented transactions. Transparency facilitates the ability of individuals with network connectivity to access and examine the data, thereby fostering accountability. The achievement of security is facilitated by using cryptographic techniques that safeguard transactions from corruption and manipulation. Simultaneously, efficiency is obtained through the implementation of a decentralized network that possesses the capability to execute transactions quickly. Traceability plays a crucial role in enabling the systematic monitoring and validation of each transaction, while programmability enables the development of decentralized apps (dApps) suited to simplifying complicated operations.

Various blockchains can be identified, namely public, private, consortium, and hybrid blockchains, each designed to cater to specific use cases. Public blockchains are characterized by their accessibility, as they are open to all individuals, and their exceptional performance in scenarios requiring decentralization, transparency, and security. On the other hand, private blockchains require limitations on accessibility, catering to a particular group, which makes them well-suited for scenarios that emphasize confidentiality and safeguarding, such as managing supply chains. Consortium blockchains are characterized by their governance structure, which involves a consortium of companies. This unique arrangement serves to bridge the difference between private and public blockchains. Hybrid blockchains combine the most-advantageous attributes of private and public blockchains, where the particular use case determines the selection of the blockchain type along with the criteria.

### 2.2. Platforms

The appearance of blockchain technology has revolutionized the data storage and transmission field, facilitating the development of secure and decentralized networks. Popular blockchain platforms, such as Ethereum, Hyperledger, and Substrate, exhibit notable differences in architectural design and distinctive features. However, they collectively provide various tools to create and implement decentralized apps (dApps).

### 2.3. Ethereum

The Ethereum platform, conceptualized by Vitalik Buterin, is an open-source and decentralized blockchain platform that aims to facilitate the development of dApps [16]. Ethereum allows programmers to design and implement smart contracts on the blockchain, facilitating the automation of complex procedures and the development of innovative dApps. Ethereum notably presents Solidity, its exclusive programming language, which serves to streamline the process of smart contract construction. The wide usage of the platform can be attributed to its adaptability, adequate security measures, and the significant size of its development and user community.

### 2.4. Hyperledger

Hyperledger, an open-source blockchain platform, is well-known for its modular architecture and is supported by The Linux Foundation. It is designed to facilitate creating and implementing customized blockchain solutions to meet the unique needs of various business use cases [17]. The emphasis placed by Hyperledger on security, privacy, scalability, and its capacity to accommodate consortium or private blockchains has rendered it a prevalent option among developers; prominent tools and frameworks within the Hyperledger ecosystem include Hyperledger Fabric, Hyperledger Sawtooth, and Hyperledger Indy.

### 2.5. Substrate

The Substrate platform, created by the Web3 Foundation and associated with Polkadot, provides a flexible framework for constructing blockchain apps [18]. The system includes a modular architectural design, including integrated tools and libraries, simplifying the development and deployment of personalized blockchain solutions for developers. The

Substrate platform's dynamic developer community appeals to anyone with a passion for designing blockchain applications that are scalable, safe, and capable of interoperability.

The Substrate framework utilizes a crucial element known as a "pallet" to facilitate the execution of logic during runtime and develop blockchain apps. Pallets can be compared to construction elements that can be assembled to develop customized blockchain applications for distinct use cases. Pallets, constructed using the programming language Rust, possess the qualities of reusability and composability. Furthermore, each pallet exhibits a distinct array of functionalities. The Substrate framework relies on their functionality to facilitate the development of blockchain applications that are modular, adaptive, and scalable, requiring minimal programming effort from developers.

In addition, Substrate incorporates the "runtime", which plays a crucial role in establishing the regulations of the blockchain, governing user and validator activities, and determining consensus procedures. The runtime, implemented in the Rust programming language, undergoes compilation into WebAssembly bytecode and operates within the Substrate runtime environment. This modular component possesses the ability to be customized in order to fulfill the specific demands of a given blockchain application.

The core infrastructure of the Polkadot system centers on the "relaychain" and "parachain" components, designed to ensure the integrity and decentralization of data transfer and communication between different blockchain networks. In essence, a relaychain or a parachain can be described as an instance of a blockchain network. The relaychain is responsible for overseeing and coordinating the various parachains within a shared network. Before being added to the blockchain network, blocks of parachains must undergo validation by the relaychain. Incorporating parachains to facilitate concurrent transaction processing decreases transaction durations and improves scalability while using the security and decentralization benefits concurrently.

In addition, Substrate incorporates the Cross-Consensus Message Format (XCM), facilitating smooth communication among separate blockchain networks inside the Polkadot ecosystem. The XCM platform enables efficient and safe communication and data transfer between blockchain networks, removing restrictions that reduce collaborative efforts. In essence, the XCM facilitates the seamless interaction of blockchains, enabling them to function as a cohesive network, optimizing the exchange of resources and information.

The Substrate platform utilizes the GRANDPA consensus method, which stands for GHOST-based Recursive Ancestor Deriving Prefix Agreement. The decision method employed by GRANDPA is based on the Greedy Heaviest Observed Sub-Tree (GHOST) algorithm. This algorithm considers block weights and chain length to achieve quick convergence on a final block. The GRANDPA protocol functions in collaboration with block production, providing a consensus system that is Byzantine-fault-tolerant. This system can deal with a certain amount of hostile nodes, ensuring that the integrity of the blockchain is not compromised.

The Substrate framework also provides various JavaScript Object Notation (JSON) Remote Procedure Call (RPC) methods that can be utilized to obtain chain-related data, execute transactions, and subscribe to events. Developers can utilize these techniques to design personalized user interfaces, automate procedures, and engage with the blockchain network through diverse means.

The selection of a blockchain platform depends on the task's specific requirements, capabilities, and limitations. Ethereum, Hyperledger, and Substrate, alongside other existing blockchain systems, offer a range of tools that facilitate the building of dApps. Our system was constructed using Substrate to leverage forthcoming scalability, interoperability, and security advancements. Although Ethereum is known for its adaptability, it is incompatible with the specific features of our product. On the other hand, Hyperledger, which is designed to cater to commercial contexts, does not meet our study's goals.

*2.6. Preliminary Ideas*

Based on the identified challenges in the education domain, such as the prevalence of counterfeit certificates and the need for enhanced transparency, this study proposes a comprehensive university management system. This proposed system uses blockchain technology to integrate functionalities into the conventional management framework. Furthermore, a robust data-management system is implemented to enhance the security and privacy of the data, utilizing a hierarchical role-assignment approach. This approach will enhance the system's capacity to effectively administer, discern, and eliminate individuals who do not meet the criteria. Utilizing the Substrate structure, it could effectively leverage notable attributes such as the relaychain and parachain model to establish a system that effectively eliminates obstacles between different blockchain systems. There are no evident concerns regarding data security and privacy protection in this situation when communicating across interconnected blockchain systems within a shared network. In addition, a comparative analysis of the previous works and ours considering blockchain security, advanced security, comprehensiveness, shareability, and connectivity is presented in Table 1.

**Table 1.** A comparative analysis of the previous works and ours.

| Ref. | Blockchain Security | Advanced Security | Comprehensiveness | Shareability | Connectivity |
|------|:----:|:----:|:----:|:----:|:----:|
| Arndt et al. [10] | ✓ | ✓ | | | |
| Badyal et al. [11] | ✓ | | | ✓ | |
| Islam et al. [12] | ✓ | ✓ | | ✓ | |
| Jeong et al. [13] | ✓ | ✓ | | ✓ | |
| Karata et al. [14] | ✓ | ✓ | | | |
| Daraghmi et al. [15] | ✓ | ✓ | | ✓ | |
| Our [15] | ✓ | ✓ | ✓ | ✓ | ✓ |

## 3. Proposed Scorechain System

*3.1. Scorechain System Overview*

The high-level design of the Scorechain system is depicted in Figure 3. The model on the left side illustrates a configuration comprising a relaychain and various parachains. Each university within our system is represented by a parachain, denoted as "UNIVERSITY". The relaychain, the "ADMINISTRATOR", is the primary network manager tasked with supervising the diverse parachains inside the Scorechain system. In order to facilitate efficient functioning, the relaychain is overseen by a council consisting of university representatives and qualified professionals, including officials of the Ministry of Education.

The Scorechain system was specifically developed to accommodate a network of universities instead of a single university. The system consists of many sub-blockchain networks, each representing an individual university. The system accommodates different people with distinct functions, such as system administrators (ADMINs), system managers (MANAGERs), teachers (TEACHERs), students (STUDENTs), and other users (NORMAL USERs). The following section describes the roles mentioned above.

- **ADMIN**: ADMINs play a crucial role in validating user accounts, encompassing administrators, managers, teachers, students, and other users. Each ADMIN carefully monitors and assesses every account to determine its eligibility. Additionally, it can approve role requests submitted by users within the system. ADMINs possess the capability to modify their data in the Scorechain system.
- **MANAGER**: System managers, also referred to as MANAGERs, supervise and coordinate departmental activities within the system. Those in charge can construct, modify, and delete departments and approve or decline user requests to join or leave these departments. The manager takes responsibility for the responsibilities related

to subjects, including the creation, updating, and removal of subjects and classes. They hold the authority to allocate or withdraw TEACHERs from teaching classes and initiate or terminate the class-enrollment process. MANAGERs also have the authority to approve student data originating from external universities. These data may include scores, certificates, awards, and examination records. They also can change their data stored in the Scorechain system.

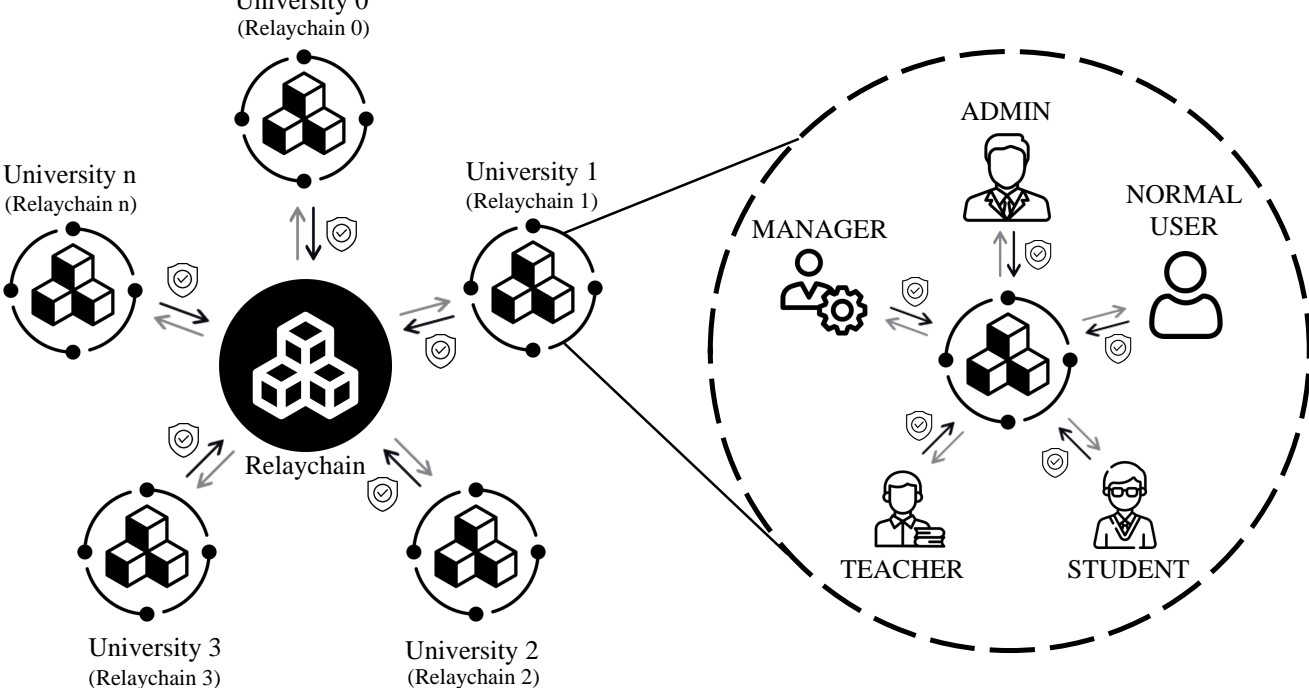

**Figure 3.** The overall architecture of the Scorechain system.

- **TEACHER**: TEACHERs can incorporate or modify educational resources intended for their assigned classes. The instructors are responsible for storing, updating, and submitting student scores for the classes under their instruction. Nevertheless, the validation of these scores is dependent on verification by MANAGERs. In addition to this, TEACHERs have the option to request membership in a specific department or make updates to their personal information through the system.
- **STUDENT**: STUDENTs can enroll in available classes, check their academic performance, modify a unique password utilized by other users to access their grades, submit requests for department membership, and change their details. Furthermore, the academic records of STUDENTs can be exported to transfer them to other universities that are part of the same network.
- **NORMAL USER**: This category encompasses individuals such as parents of students, recruiters, and other users. NORMAL USERs possess a restricted range of actions, such as the ability to modify their personal information, submit applications for positions such as ADMIN, MANAGER, TEACHER, or STUDENT by providing the required documents, and retrieve a particular student's academic record by utilizing the STUDENTs' unique password and STUDENTs' identification numbers. The process for NORMAL USERs to request roles is detailed in Figure 4.

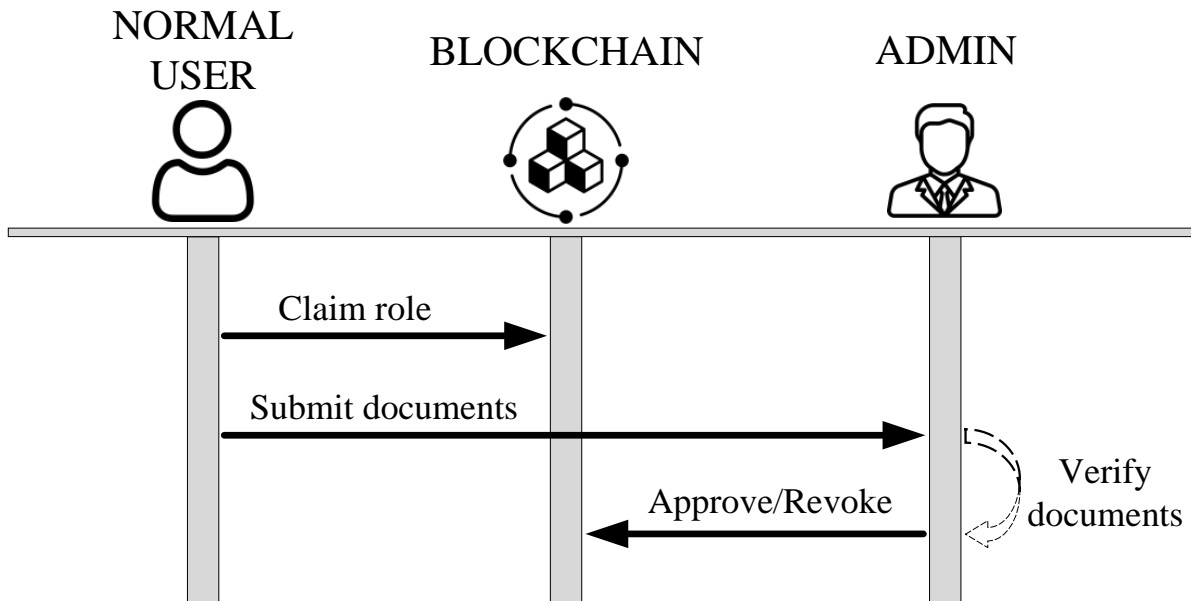

**Figure 4.** Process for role request.

*3.2. A Comprehensive University Management System*

In contrast to previous studies that have concentrated on specific aspects of student data administration on the blockchain, our proposed Scorechain system presents an extensive platform for university management. The scope of this phenomenon surpasses the mere storage and validation of discrete elements such as transcripts, certificates, or awards. In contrast, our system replicates the features of a traditional university management system, accommodating the various requirements of its users.

The system facilitates the utilization of its functions by different user roles, including ADMINs, MANAGERs, TEACHERs, STUDENTs, and NORMAL USERs. These functionalities are comparable to those often found in a conventional university administration system. The functions described comprise the processes involved in managing departments, subjects, and classes. Furthermore, the system is crucial in organizing and controlling student-related information, including scores, certificates, and examinations. Moreover, any standard user within the Scorechain system, including recruiters, parents of students, other educational institutions, or other users, can access students' information, depending on the owner of such information, allowing them authorization to do so. Our services encompass an account management system incorporating an approval/revocation mechanism, which will be elaborated upon in Section 3.3. Furthermore, the secure and private transfer of students' information within the Scorechain network, specifically in the context of multi-university management systems, will be demonstrated in Section 3.4. Figure 5 presents a block diagram that showcases the potential evolution of the proposed Scorechain system into a comprehensive university administration system.

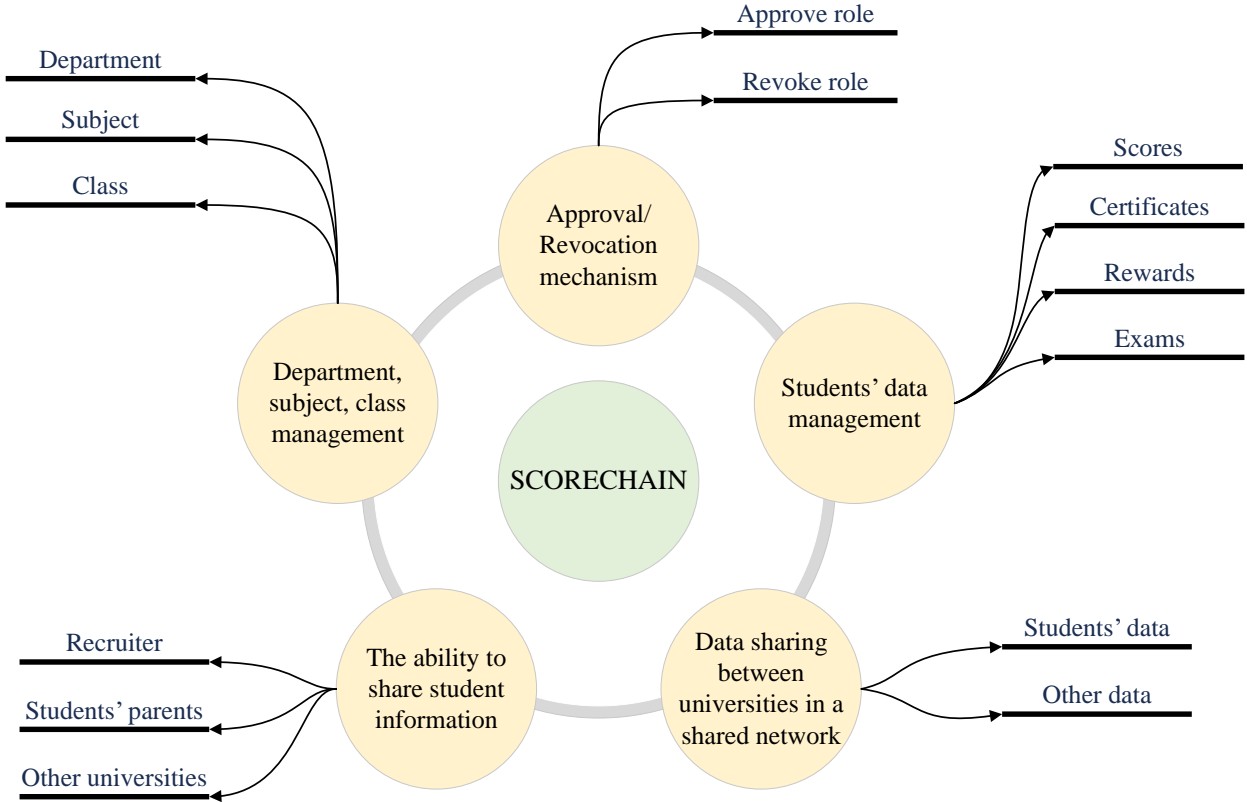

**Figure 5.** Comprehensive university management system.

### 3.3. Highly Secure Data Management Through a Hierarchical Role Assignment Mechanism

The main objective of this research is to create a comprehensive university management system that goes beyond the traditional function of storing and managing student data within a blockchain ledger. The main goal in this context is to prioritize preserving data privacy and to take proactive measures to fix security vulnerabilities that could result in unauthorized data exposure. In order to enhance the security infrastructure of the Scorechain system, this study suggests incorporating two hierarchical management systems within the Scorechain platform. A commonly seen issue in such systems involves withdrawing privileges or advantages from individuals who demonstrate poor performance or fail to satisfy qualifying criteria. On the other hand, persons who satisfy the qualifications for particular responsibilities are authorized to fill these positions. In order to enhance operational efficiency and mitigate the burden on individual management administrators or managers, the allocation of functions and responsibilities is divided among several qualified groups.

The admin hierarchy system, as depicted in Figure 6, is responsible for overseeing all ADMINs within our system. The admin hierarchy system comprises three hierarchical layers, denoted as Layers 0, 1 and 2. The Layer 0 ROOT ADMIN possesses the highest level of authority. The ROOT ADMIN possesses the authority to allow or remove role permissions for the ROOT MANAGER (described in the following paragraph). The ADMINs in Layer 1 have the authority to approve and revoke permission for MANAGERs in Layer 1 within the manager hierarchy system. This process is repeated in Layer 2 as well. In our system, all ADMINs also have the authority to give and take away responsibilities from STUDENTs and TEACHERs.

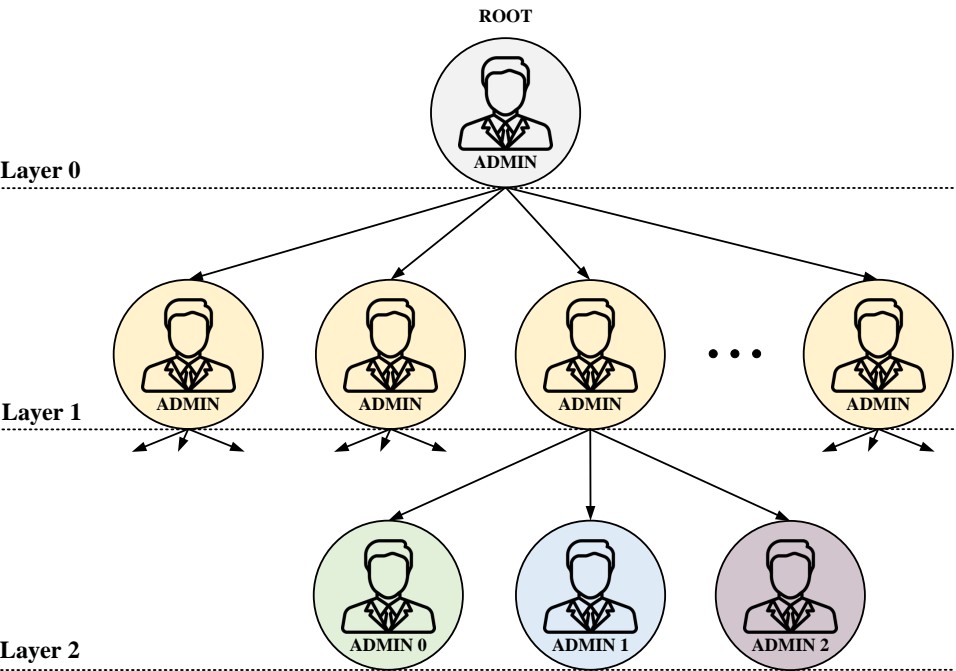

**Figure 6.** Hierarchical admin management.

Furthermore, the manager hierarchy system is shown in Figure 7. At Layer 0, the ROOT MANAGER has the exclusive authority to add, modify, and remove departments. The individuals occupying managerial positions in Layer 1 serve as department heads and are appointed by the ROOT MANAGER. They can create, modify, and delete assignments related to various classes and subjects. Each MANAGER in Layer 1 oversees three subordinates, each assigned specific responsibilities. MANAGER 0 can add and remove people from his/her departmental membership. MANAGER 1 covers the assignment and removal of teachers and the facilitation of enrollment for classes opening and closing. MANAGER 2 is responsible for approving students' scores, certificates, awards, and examinations.

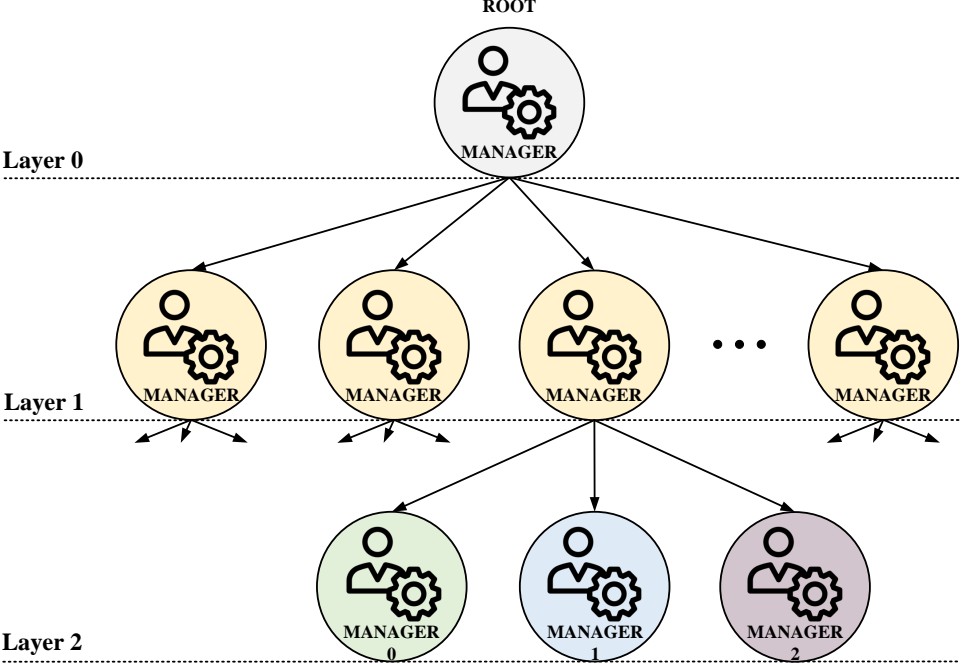

**Figure 7.** Hierarchical manager management.

### 3.4. Multiple Universities' Highly Secure Privacy and Data-Sharing Platform

In the realm of incorporating conventional blockchain systems in the educational sector, an essential and frequent obstacle emerges regarding the intercommunication between these independent systems. The current concept of the design of blockchain systems is fundamentally based on the principle of independent operation. Therefore, the need for defined communication routes between these separate blockchain systems and the methods used to transmit information can lead to severe security and privacy concerns.

The difficulty at hand is effectively tackled by the Scorechain system, which achieves this by creating a comprehensive network that incorporates various blockchain networks. This network consists of a relaychain and parachains, assisted by the Substrate blockchain framework. The Scorechain system utilizes individual blockchain systems to effectively and securely exchange data within the network. This approach is achieved by employing the XCM, as explained in Section 2.5. The relaychain and parachain structure effectively manages the communication between blockchain systems operating within a shared network, ensuring continuous data transmission while maintaining high security and anonymity. Furthermore, the Scorechain system exhibits intrinsic scalability due to its architectural design consisting of a relaychain and parachains.

## 4. System Implementation

### 4.1. A Comprehensive Implementation

As depicted in Figure 8, the system's full implementation comprises three fundamental components: the Front-End, the Back-End, and the Blockchain. Every component within the Scorechain system fulfills a unique purpose in its operation.

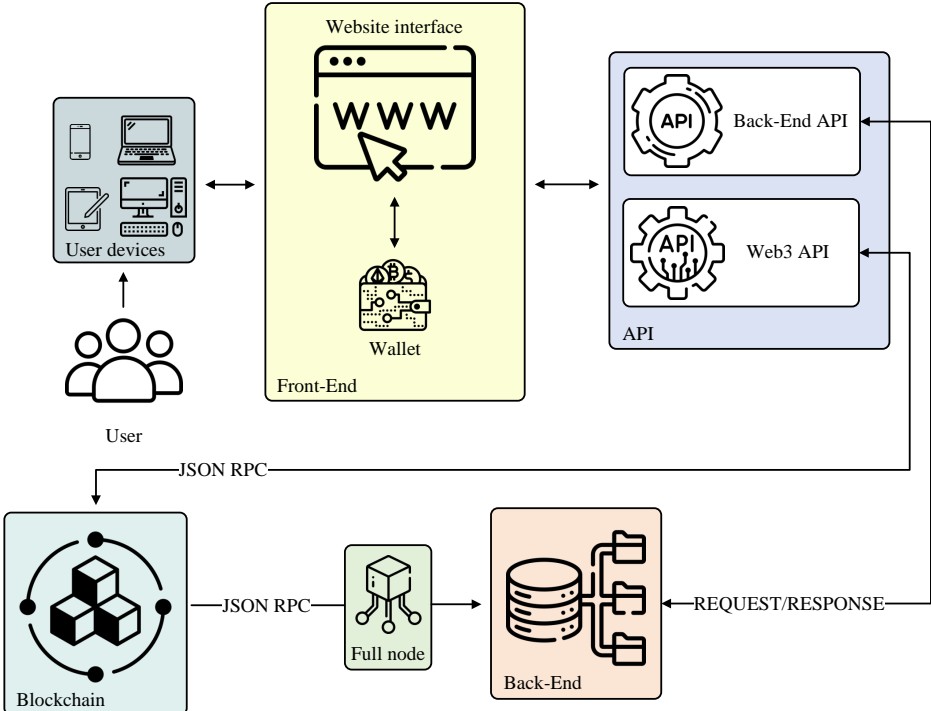

**Figure 8.** A comprehensive view of the implementation system.

The Front-End component is responsible for developing a user-friendly web interface, improving the overall user experience, and facilitating smooth interaction with the system. The provided interface facilitates user interaction with the blockchain component through Polkadot's blockchain Application Programming Interface (API) while depending on a Back-End API to handle information acquisition and retrieval from the Back-End system.

The utilization of ReactJs, a widely recognized JavaScript package, plays a significant role in advancing the Front-End, particularly in constructing interactive user interfaces.

The Back-End system is an off-chain storage mechanism kept within a database and assisted by a listener operating on a full node. This approach provides enhanced efficacy in the retrieving and extracting of data in comparison to a system based on blockchain technology. Within the present framework, the Back-End system is responsible for storing and monitoring user account data, encompassing the procedures of user registration and login procedures. The Back-End system is implemented using Typescript, a programming language derived from JavaScript and created and maintained by Microsoft.

The Blockchain is the essential element of the Scorechain system, functioning as the storage platform for the Scorechain ledger. The present implementation is in adherence to the principles discussed in Sections 3.2–3.4, namely the comprehensive university management system, the hierarchical role-assignment mechanism, and the secure data-sharing platform. The Substrate framework facilitates interactions with the blockchain by employing standardized protocols such as the JSON-RPC protocol. These protocols facilitate the effective interaction of developers with the blockchain network across the Internet.

### 4.2. Implementation of the Comprehensive University Management System

Implementing the comprehensive university management system involves using pallets, as illustrated in Figure 9. The system is supported by three main pallets: the Account pallet, the Subject pallet, and the Department pallet. Each pallet consists of functions, also known as extrinsic calls, that can be accessed by various users, as explained in Section 3.1.

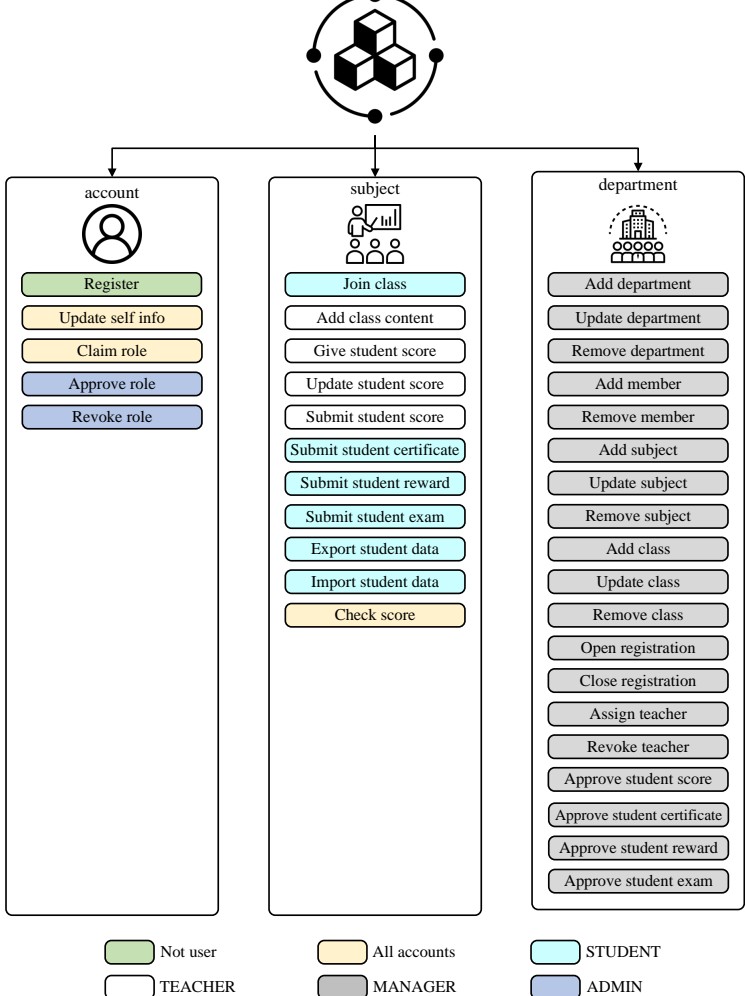

**Figure 9.** A basic illustration of the pallets and functions of the proposed system.

- Account pallet: With the *Register* function, people can join the Scorechain system as NORMAL USERs. *Update self info* will help all users update their information. *Claim role* lets users in our system claim another role. *Approve role* and *Revoke role* ADMINs approve a role requested by a user and revoke a role from a user if he or she is no longer eligible.
- Subject pallet: First, *Join class* will let STUDENTs join a class that is opening for registration. Second, the *Add class content* function lets TEACHERs who are assigned to a class add the content and material for that class. Next, *Give score*, *Update score*, and *Submit score* can be used by the TEACHERs of a class to give a score, update scores for STUDENTs, and submit scores to a specific MANAGER to approve. In addition, STUDENTs can use *Submit certificate*, *Submit reward*, and *Submit exam* to submit students' data to be approved by a MANAGER.
- Department pallet: First of all, *Add department*, *Update department*, and *Remove department* can help MANAGERs manage departments in our system. Next, the MANAGERs of a department can add or remove members from their department. The MANAGERs can create, update, and remove subjects and classes through *Add subject*, *Update subject*, *Remove subject*, *Add class*, *Update class*, and *Remove class* functions. In addition, MANAGERs can open or close a class registration through *Open registration* and *Close registration*. TEACHERs can be assigned to or revoked from a class. Furthermore, MANAGERs can approve students' data such as scores, certificates, rewards, and exams through *Approve score*, *Approve certificate*, *Approve reward*, and *Approve exam*.

### 4.3. Implementation of the Hierarchical Role-Assignment Mechanism

We implemented the hierarchical role-assignment mechanism (as presented in Section 3.3) by three functions of pallet *account*, *Claim role*, *Approve role*, and *Revoke role*. In addition, two parameters, *role* and *role_status*, are used to indicate an account's status. Several steps must be completed to successfully claim a role, as described in Figure 4:

- Step 1: When the *Claim role* function is called, the account ID can be obtained from the signature on the transaction to be sent. The account information stored on the blockchain will be easy to obtain using the account ID. After that, the value of *role* is changed to the role that the accounts claimed, such as STUDENT, TEACHER, MANAGER, or ADMIN. Moreover, *role_status* is changed to *Pending*, which means the role is being requested. In parallel, some documents must be sent to the ADMIN in an off-chain service (website interface).
- Step 2: When the *Claim role* function is executed, ADMINs will use the *Approve role* and *Revoke role* to decide whether the user, who executed the *Claim role* function, can obtain the new role or not depending on his/her submitted documents. If *Approve role* is called, the new status of *role_status* will change to *Approved*. In contrast, if *Revoke role* is called, the status of *role_status* and *role* will be changed to initial.

In addition, if a user's performance is not good, he or she can also be revoked from our system through *Revoke role* by ADMINs.

### 4.4. Implementation of the Multiple Universities' Highly Secure Privacy and Data-Sharing Platform

The Scorechain system we propose has two main types of nodes: collator nodes and validator nodes. First, in the context of a parachain, the primary responsibility of collator nodes is to collect transactions and produce blocks specific to the given parachain. Furthermore, validator nodes assume an essential function in safeguarding the security and preserving the integrity of the network. Within the framework of parachains, validator nodes validate and ultimately affirm the authenticity of blocks produced by collator nodes. Second, the validator nodes on the relaychain are entrusted with the responsibility of verifying and ultimately approving the blocks generated by the parachain's collator and validator nodes themselves. The complete list of procedures encompassed in the production and finalization of parachain blocks is depicted in Figure 10.

- Step 1: In a parachain, collators produce blocks by gathering transactions submitted to the parachain.
- Step 2: After a block is created, it is validated by the parachain's validator set, which includes collators, fishers, and other node kinds, in addition to regular validators.
- Step 3: If the parachain's validator set deems the block legitimate, it is sent to the relaychain, which is validated by the relaychain's validator set.
- Step 4: If the relaychain's validator set validates the block and overcomes the consensus, a relaychain block will be created and added to the relaychain's blockchain.

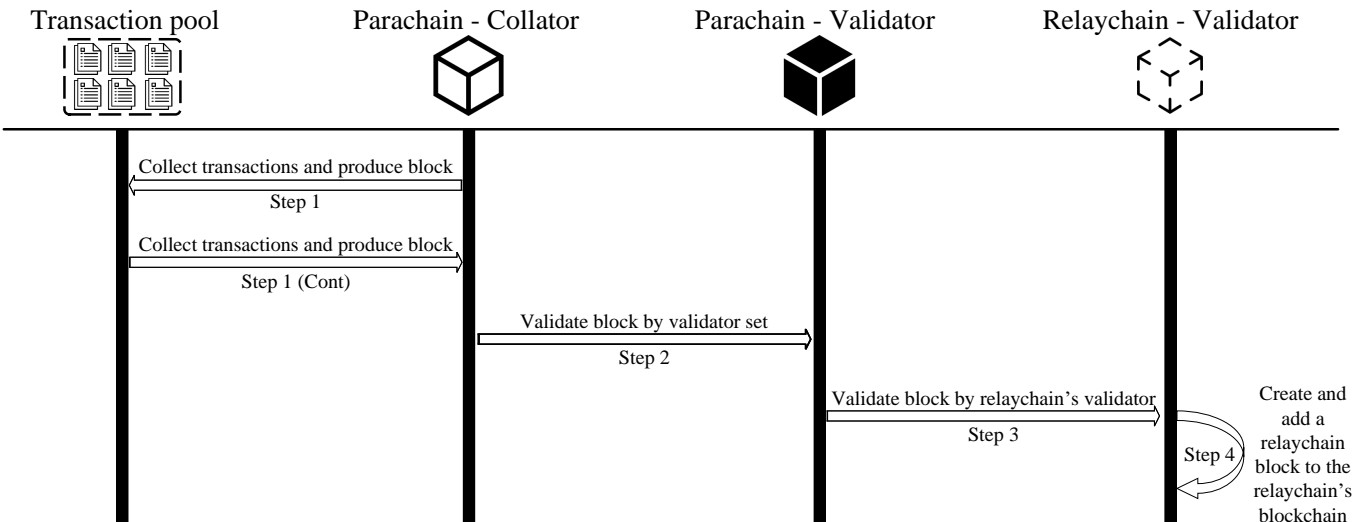

**Figure 10.** Four steps from producing to finalizing a block in a relaychain and parachain model.

## 5. Discussion

To establish a sustainable blockchain system for education, Scorechain integrates three fundamental capabilities to guarantee student data's convenience, privacy, and security. This analysis will examine the methods by which Scorechain accomplishes these objectives.

- *Resource management*: Scorechain is a comprehensive system specifically developed to store and effectively manage a wide range of student information, encompassing scores, awards, certificates, and examinations. This approach facilitates the ability of students, parents, and recruiters to access and monitor student progress and accomplishments conveniently. Furthermore, in our pursuit to establish a system based on blockchain technology for educational institutions, Scorechain facilitates the effective administration of subjects, departments, and human resources. The Scorechain system ensures sustainable and reliable resource management by utilizing the various features of blockchain, including decentralization, immutability, transparency, security, efficiency, traceability, and programmability.
- *Hierarchical employee management system*: Implementing a hierarchical employee management system in Scorechain is paramount in bolstering the system's security and reliability. The present system can grant or withdraw authorization for accounts within our network. Through its oversight, we ensure the safety and accuracy of our system. Including a hierarchical employee management system is crucial in enhancing the overall security and reliability of the Scorechain network, aligning with our sustainability objectives.
- *Multi-university management system*: The multi-university management system developed by Scorechain enables smooth communication and integration between participating institutions by utilizing the relaychain and parachains within the Substrate framework. Establishing interconnectivity enables universities to communicate more safely while safeguarding privacy and security considerations. As a result, the transfer

of student data between universities can be carried out efficiently and securely. The Scorechain system enhances the functionalities of Scorechain beyond the conventional blockchain network while upholding the fundamental characteristics of blockchain technology. This approach enables us to advance sustainability and cultivate collaboration within the educational ecosystem.

In addition, we conducted an assessment and compilation of the duration required for storing and retrieving student records. This evaluation was performed on datasets containing 100, 200, 300, 400, 500, 600, 700, 800, 900, and 1000 existing student records, following the prescribed guidelines outlined in the official documentation [19]. The experiment was conducted on a system running Ubuntu 22.04.2, equipped with an Intel Core i9-12900 processor featuring a 30M cache and a clock speed of up to 5.10 GHz. The findings are presented in Figure 11. Subsequently, the patterns exhibited by two distinct categories of columns exhibit dissimilarities. As illustrated in Figure 11, the time needed to store a student record exhibits a substantial linear growth pattern with increased student records. In contrast, the duration needed to retrieve a student record exhibits a linear growth pattern, albeit not as pronounced as the column above category. Consequently, it can be inferred that there exists a positive correlation between the quantity of student records within our system and the duration required for data storage. Moreover, the quantity of student records minimally influences the duration required to retrieve data through querying. In addition, no existing works share the same platform and objective as ours, thereby constraining our ability to provide a comprehensive comparison.

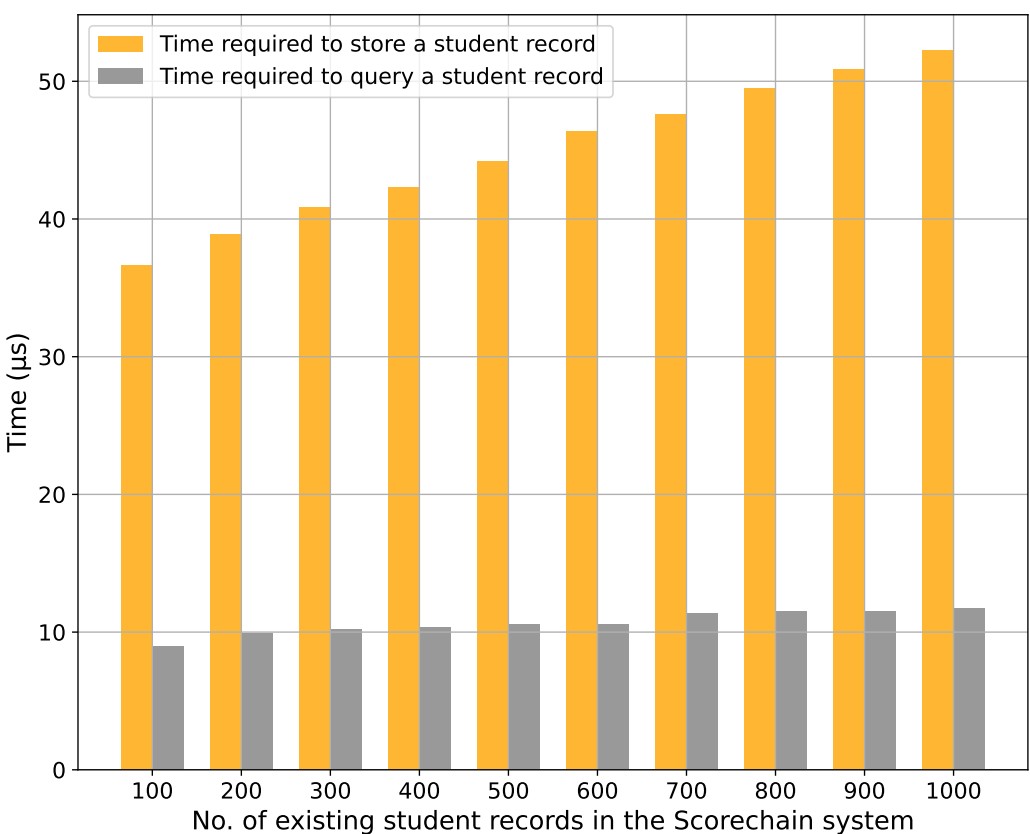

**Figure 11.** Time taken to store and obtain a student record through the Scorechain system.

## 6. Conclusions

This study presented the development of a blockchain-based system designed to manage student data within education. The proposed system integrates three essential components to improve efficacy, security, and privacy, establishing a sustainable solution. The successful performance of the Scorechain system was demonstrated through com-

prehensive testing. Additionally, a hierarchical system for employee management was implemented, resulting in enhanced operational efficiency. Finally, the Scorechain system guarantees uninterrupted communication with other network participants, protecting privacy and enhancing security. Even with these accomplishments, specific limitations necessitate attention in subsequent assessments. Furthermore, the proposition outlined in this paper entails developing and investigating a robust privacy and data-sharing platform tailored to the needs of multiple universities. Potential future directions for our system encompass improving its user-friendliness, establishing a privacy and data-sharing platform that caters to multiple universities, and prioritizing refining the consensus algorithm within our blockchain network. It is imperative to recognize that the assessment and juxtaposition of our research are limited by certain factors, necessitating further progress.

**Author Contributions:** Investigation, V.D.T.; methodology, T.H.T., S.A., D.K.L., and H.L.P.; project administration, T.H.T.; supervision, T.H.T.; writing—original draft, V.D.T.; writing—review and editing, T.H.T. All authors have read and agreed to the published version of the manuscript.

**Funding:** This work was supported by the Japan Science and Technology Agency (JST) under a Strategic Basic Research Programs Precursory Research for Embryonic Science and Technology (PRESTO) under Grant JPMJPR20M6.

**Institutional Review Board Statement:** Not applicable.

**Informed Consent Statement:** Not applicable.

**Data Availability Statement:** Not applicable.

**Conflicts of Interest:** The authors declare no conflict of interest.

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
