# Peer review of "Blockchain-Powered Education: A Sustainable Approach for Secured and Connected University Systems"

_sustainability, doi:10.3390/su152115545_

Round 1
Reviewer 1 Report
This is not a scholarly article but the presentation of a product designed by the authors.
There are many places with the wrong choice of words/terms.
Reviewer 2 Report
Important study. Very clearly and methodically presented. However in-text referencing should be corrected as commented on the paper.

Minor editing in writing is needed (i have marked many places on the paper).
Reviewer 3 Report
Dear Author/s
Please refer to the document attached for further suggestions and comments.
Best Regards

Reviewer 4 Report
The article is very interesting and shows remarkable entrepreneurship on the part of the authors, who are carrying out a technology transfer process that deserves discussion and testing by educational institutions and systems.
It is very well structured, and the presentation and argumentation is of high quality and very rigorous.
It is well grounded in theory and supported by appropriate bibliography.
The graphic quality of the illustration of how Scorechain works is also noteworthy.
The discussion it develops about the benefits of blockchain falls within the scope of believers and converts to the technology, which is done very proficiently and effectively.
The benefits of the technology - ergo, of Scorechain - are understood as generating efficiency, security, integration and interoperability, and practice and use are its source of endorsement.
The use of these systems must always be careful of the dystopia that is always hanging over societies, as today's war in Ukraine constantly reminds us. The immense energy resources needed to run technological systems are not always guaranteed and contingencies must be taken into account when managing an immense mass of data that reflects the lives of the people who make up the institutions.
Finally, people's participation must always be taken into account: the temptation to develop systems that dispense with them and determine their lives is a constant and the verified results must not be forgotten.
Congratulations to the authors on their excellent work and all the best for the article!
Reviewer 5 Report
First, I appreciate the authors developing this study on blockchain-based education.
However, as you have mentioned it as a sustainable approach, please explain how and to what extent.
For further relevance, in the introduction section, please cite the following paper recently published in Sustainability.
Paper link: https://www.mdpi.com/2071-1050/15/2/1470
Paper title: Blockchain in Online Learning: A Systematic Review and Bibliographic Visualization
Moderate editing of the English language is required.
Round 2
Reviewer 1 Report
The authors have clearly made an effort to address the recommendations and observations of the reviewers. Unfortunately, the article is not improved, quite the contrary. There are very serious issues of academic writing that need to be better addressed. These include many remaining and new basic grammar errors and almost meaningless sentences. In their reply to the reviewers, the authors correctly indicate that the draft includes sections with significant academic and professional substance. Yet, the nature of the article has not changed: this is not a scholarly article but the presentation of a product developed by the authors. If the Journal wishes to publish presentations of products, this article would qualify. provided writing standards are met.
Significant re-writing needed
Round 3
Reviewer 1 Report
Language and writing issues raised previously have been addressed.